# Excitation-Dependent Fluorescence Helps to Indicate Fungal Contamination of Aquatic Environments and to Differentiate Filamentous Fungi

**Elena Fedoseeva** [1,*]**, Svetlana Patsaeva** [2] **, Devard Stom** [3,4,5] **and Vera Terekhova** [1,6]

1   Laboratory of Ecological Functions of Soil, Severtsov Institute of Ecology and Evolution, Russian Academy of Sciences, 119071 Moscow, Russia
2   Faculty of Physics, Lomonosov Moscow State University, 119991 Moscow, Russia
3   Faculty of Biology and Soil, Irkutsk State University, 664003 Irkutsk, Russia
4   Baikal Museum of the Siberian Branch of the Russian Academy of Sciences, 664520 Listvyanka, Russia
5   School of Architecture, Construction and Design, Irkutsk National Research Technical University, 664074 Irkutsk, Russia
6   Faculty of Soil Science, Lomonosov Moscow State University, 119991 Moscow, Russia
*   Correspondence: fedoseeva@sev-in.ru; Tel.: +7-925-625-43-51

**Abstract:** Fungal contamination of aquatic environments can lead to an adverse impact on the environment and human health. (1) The search for fast, inexpensive and appropriate methods for detection of fungi is very moving rapidly due to their significant impact on ecosystem functions and human health. (2) We focused on examination of fluorescence proxies able to distinguish chromophoric matter occurring in different fungi. Spectroscopic studies were performed on five strains of filamentous fungi: *Trichoderma harzianum, Fusarium solani, Alternaria alternata, Cladosporium cladosporioides* and *Aspergillus terreus*. (3) The results showed that most of the fungal autofluorescence was emitted by amino acids, melanin-like compounds, NAD(P)H and flavins. The spectra of five fungal species cultivated as planktonic or surface-associated forms turned out to be different. Protein fluorescence can be used to detect general microbial contamination. Presence of excitation wavelength dependent mode and the "blue shift" of fluorescence (emission bands 400–500 nm) can be suggested as specific feature of fluorescence of fungal melanin-containing samples. (4) The determination based on fluorescence spectra obtained at a certain excitation/emission wavelengths pair and at whole excitation-emission matrices (EEMs) coupled to principal component analysis (PCA) algorithms as a tool of improving detection capabilities can be suggested to enable fast and inexpensive monitoring of fungal contamination of aquatic environments.

**Keywords:** fungal contamination; excitation-dependent fluorescence; excitation/emission matrix

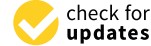



## 1. Introduction

Fungi, being an integral component of ecosystems, control a wide range of biosphere functions [1]. Fungi can be found almost everywhere in the ecosystems, and are able to grow in a wide variety of aquatic environments [1,2]. Mycelial forms (filamentous fungi) predominate among the aquatic fungi, in particular the many species from genera (*Aspergillus, Alternaria, Penicillium, Trichoderma, Cladosporium, Fusarium* and some others) [3,4]. Aquatic fungi can have a substantial environmental impact by entering into a symbiotic relationship, acting as infectious agents for hydrobionts, and causing many human health problems [3,5]. Fungal contamination can lead to an adverse impact on the environment, as well as on the human health. The role of fungi becomes more important in coastal zones. Aquatic substrates, such as algae, plants and plant residues and other sources of allochthonous organic matter (OM), concentrated mainly along the coastline provide the conditions for fungi growing [3]. Aquatic fungi include two groups: water-borne and re-released to aquatic environments from terrestrial habitats [2,6]. In terrestrial and aquatic ecosystems, fungi

can serve as indicators of the ecological state and dynamics of OM. Pollutions and other destructive environmental processes lead to negative structural and functional deformation of fungal communities [2,7,8].

Fungi are known as producers of a large set of primary and secondary metabolites. Primary metabolites are indispensable products of metabolic processes, namely, proteins, carbons, lipids, nucleic acids and coenzymes [9,10]. Secondary metabolites include toxins and pigments of different nature, in particular, carotenoids, quinones and melanins, causing pigmentation of fungi. Fungi synthesize a wide range of toxins: antibiotics, phytotoxins, mycotoxins and complex action toxins, dangerous to animals and humans [11]. The degree of pathogenicity of the species is determined by its ability to synthesize both toxins and melanins [2].

A number of laboratory techniques were developed for detection of fungal biomass and processes occurring with fungal participation in aquatic ecosystems. Among them one can find the microbiological cultural, molecular-genetics (PCR detection and sequencing) and chromatographic methods [5,12,13]. Detection of fungi in aquatic ecosystems using traditional microbiological, molecular-genetics and chromatographic methods is species-specific, as well as time-consuming and instrument-complicated. Over the last few decades, there have been many investigations on fungi, aimed to monitor bioprocesses with the help of optical technologies, such as in situ microscopy, UV-visible, near-infrared and Raman spectroscopy. Optical methods are most widely used to detect airborne fungal particles [14–16]. Interesting image techniques have been developed for fungal spores monitoring: diffraction fingerprint image acquisition [17]. Optical detection of fungi in aquatic environments due to its specifics has very limited applications, especially performed on mycelial forms of fungi (filamentous fungi) [18–21]. It seems relevant to search for new methods to detect fungi in water, and spectral methods could provide us with fast and non-destructive methods in a non-contact mode. Fluorescence spectroscopy has been shown to be an important tool to characterize various OM in water sources [22]. Fluorescence was linked to water quality indicators, such as pigments of anoxygenic phototrophic bacteria [23], microbial biomass and, consequently, microbial contamination [24,25].

Our research was focused on the examination of autofluorescence of five strains of filamentous fungi with different pigmentation linked to melanin formation in the species. Our objectives were: (1) to investigate features of excitation-dependent fluorescence emissions of filamentous fungi cultivated in different conditions; (2) to assess the possibility to quantify fungal biomass and to discriminate fungal samples based on fluorescence spectral features; and (3) to discuss potential applications of these results for environmental assessment and management.

## 2. Materials and Methods

The microscopic filamentous fungal strains *Trichoderma harzianum* Rifai, *Alternaria alternata* (Fr.) Keissl, *Cladosporium cladosporioides* (Fresen.) G.A. de Vries, *Fusarium solani* (Mart.) and *Aspergillus terreus* Thom isolated from soil were investigated in laboratory conditions (photographs of fungal colonies are presented in Figure 1). These species can be considered as amphibious species for aquatic environments. *T. harzianum* does not produce melanins. *C. cladosporioides* and *A. alternata* are able to synthetize 1,8-dihydroxynaphthalene (DHN-) melanin-like compounds that have been mostly associated with ascomycetes [26]. Representatives of the *Fusarium* and *Aspergillus* genus are capable of synthesizing melanins less canonical for ascomycetes—5-deoxybostrycoidein-melanin and Asp-melanin, respectively.

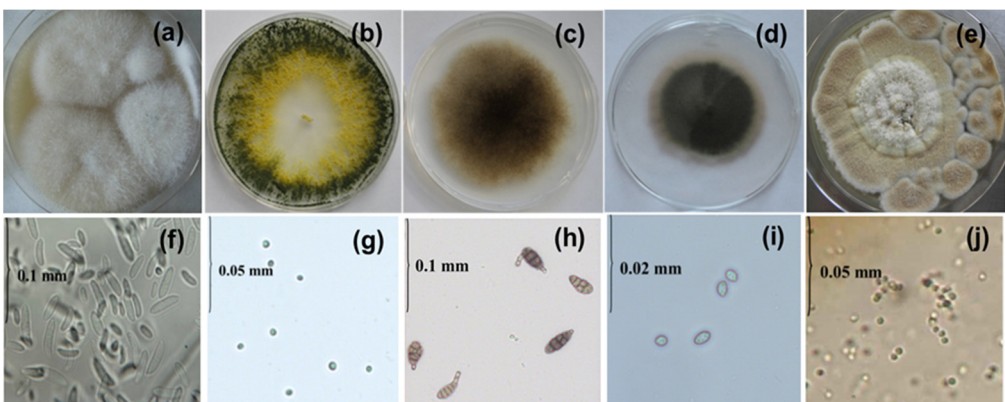

**Figure 1.** Photographs of spore-bearing colonies on agar medium (**a**–**e**) and spores (conidia) under a microscope (**f**–**j**): (**a**,**f**) *Fusarium solani*; (**b**,**g**) *Trichoderma harzianum*; (**c**,**h**) *Alternaria alternata*; (**d**,**i**) *Cladosporium cladosporioides*; (**e**,**j**) *Aspergillus terreus*.

Fungi were cultivated at 22 °C on Czapek nutrient medium with the mineral composition (g $L^{-1}$): $NaNO_3$—3.0, $K_2HPO_4$—1.0, $MgSO_4$—0.5, KCl—0.5, $FeSO_4$—0.001, pH 5.5–6.0. As a carbon source, 30 g $L^{-1}$ sucrose was added. Two different cultivation methods were used: planktonic culture on liquid medium, LM, (without agar) and surface-associated culture on containing agar medium (15 g $L^{-1}$). In the first method, fungi were cultivated in a submerged culture using a series of 250 mL of sterilized flasks containing 100 mL of nutrient medium. Inoculation of fungi into the flasks was performed with addition of spore suspension giving an average count of $1*10^5$–$10^6$ spores/mL. The number of conidia was counted in a Goryaev counting chamber of 0.9 $mm^3$ volumes. The inoculated flasks were incubated at 24 ± 2 °C for 5–7 days with constant shaking at a 150 rpm/min before the formation of fungal non-spore mycelium pellets. For spectral measurements the supernatant liquid was filtered with membrane filters to remove mycelium particles, and only extracellular metabolites were analyzed. In the second method, fungal mycelium was cultivated on Petri dishes with agar-containing medium (AM). Inoculation was carried out by transferring fungal spores from the stock fungal culture. Petri dishes were kept in a climate control box at 24 ± 2 °C for 5–7 days until formation of spore-bearing colonies. For spectral measurements the suspensions of fungal spores in water were prepared.

Fluorescence emission spectra were measured using a Solar CM2203 luminescence spectrometer with excitation radiation at several wavelengths (280, 310, and 370 nm) for aqueous samples placed in quartz cuvettes. Fluorescence excitation spectra were detected for protein fluorescence at 350 nm and for other fluorophores at 440 nm. For all investigated fungal samples we calculated fluorescence quantum yield (QY) that serves as an intrinsic property for the characterization of individual fluorophores and their interaction with environment. QY values were calculated based on both the wavelength-integrated fluorescence intensities and absorbance values at an excitation wavelength (measured for diluted samples) as described in [27]. To create excitation emission matrix (EEM) that is suitable for identifying and quantifying components in complex samples fluorescence emission spectra were recorded every 10 nm between 250 and 650 nm [7,8,28].

The experiments were carried out in 3–5-fold biological replications. For each biological replication, a culture liquid was sampled, or an aqueous suspension was prepared for spectral measurements twice. Principal component analysis (PCA) was carried using XLSTAT Software. We incorporated into PCA normalized fluorescence intensity in the 290–650 nm region upon 280 nm excitation, in the 350–650 nm region upon 310 nm excitation, and in the 400–650 nm region upon 370 nm excitation for five fungal strains. The EEM spectral images of the fungal samples were carried using Origin Software (OriginPro 8.1 SR2, OriginLab Corporation, Northampton, MA 01060, USA).

## 3. Results

### 3.1. Fluorescence Excitation/Emission Matrix (EEM) Spectroscopy

In recent years, EEM fluorescence spectroscopy, being an easy to implement, fast and relatively not so expensive method, has been widely used for qualitative characterization of dissolved OM of various origins [29], as a potential rapid and simple method for bacterial detection in water [15], to monitor OM dynamics in the processes of microbial bioremediation of wastewaters, leachate, sewage [8,30].

The EEM spectra of culture liquids *A. alternata*, *C. cladosporioides*, *F. solani* and *T. harzianum* have similar emission regions at 300–370 nm upon excitation at 250–300 nm that can correspond to the fluorescence of proteins [31,32] (Figure 2a–d). The EEM images did not distinguish such a protein-like fluorescence for *A. terreus*; the fluorescence emission is located in 350–500 nm with excitation at 250–270 nm (Figure 2e). The spectral bands corresponding to the fluorescence of melanins and NAD(P)H differs in the studied samples. It is not observable on the EEM of the *F. solani* culture (Figure 2c). On the EEM of *T. harzianum* culture, this emission was found at 370–500 nm when excited at 250–350 nm (Figure 2d). The most similar are the spectral regions of emission for the cultures *A. alternata* and *C. cladosporioides*: fluorescence at 400–470 nm with excitation at 320–350 nm (Figure 2a,b). This fluorescence is usually associated to lignocellulose like compound, humic acid-like or melanoidin-like compound fluorescence [31]. The EEM of *A. terreus* culture deviates from others by the presence of an abnormally wide emission band spreading from 350 nm to 550 nm upon excitation with wavelengths at intervals of 300–450 nm and indicating the presence of unique composition of chromophores of *A. terreus* (Figure 2e). *A. terreus* is a producer of many aromatic secondary metabolites. The main secondary metabolites are various derivatives of asperlides and aspernolides, butyrolactones and butenolides and terrusnolides, which can potentially be the source of the unique composition of chromophores of *A. terreus* [33].

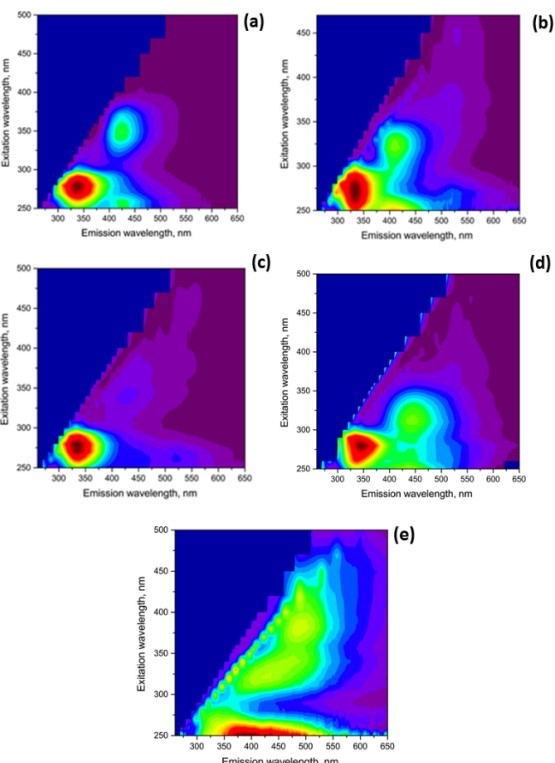

**Figure 2.** Excitation-emission matrix (EEM) of the fungal samples: (**a**) *Alternaria alternata*, (**b**) *Cladosporium cladosporioides*, (**c**) *Fusarium solani*, (**d**) *Trichoderma harzianum*, and (**e**) *Aspergillus terreus*. Note: the color reflects the fluorescence intensity. The fluorescence intensity decreases in the order of colors: burgundy—red - yellow - green - blue - dark blue.

### 3.2. Fluorescence Spectral Features Analysis of Fungi Samples Cultivated on Agar-Containing Medium

Typical autofluorescence of fungi spores suspended in water under excitation at 280 nm consist of two overlapping bands: the UV emission band with a maximum at 300–350 nm, protein fluorescence, and the broad band in the blue region with a maximum at 400–500 nm, resembling the fluorescence of NAD(P)H or melanin-like compounds (Figure 3c–e). The maximum in the excitation spectra for the protein emission band was found at 280 nm, which additionally proves the protein-like or phenolic-like origin of the fluorophores emitting at 350 nm (Figure 3a). It is clearly seen that the excitation spectra for protein fluorescence registered at 350 nm are identical for the four fungal strains. An exception is the *A. terreus* strain with emission maximum located in the region of 320 nm upon excitation at 280 nm.

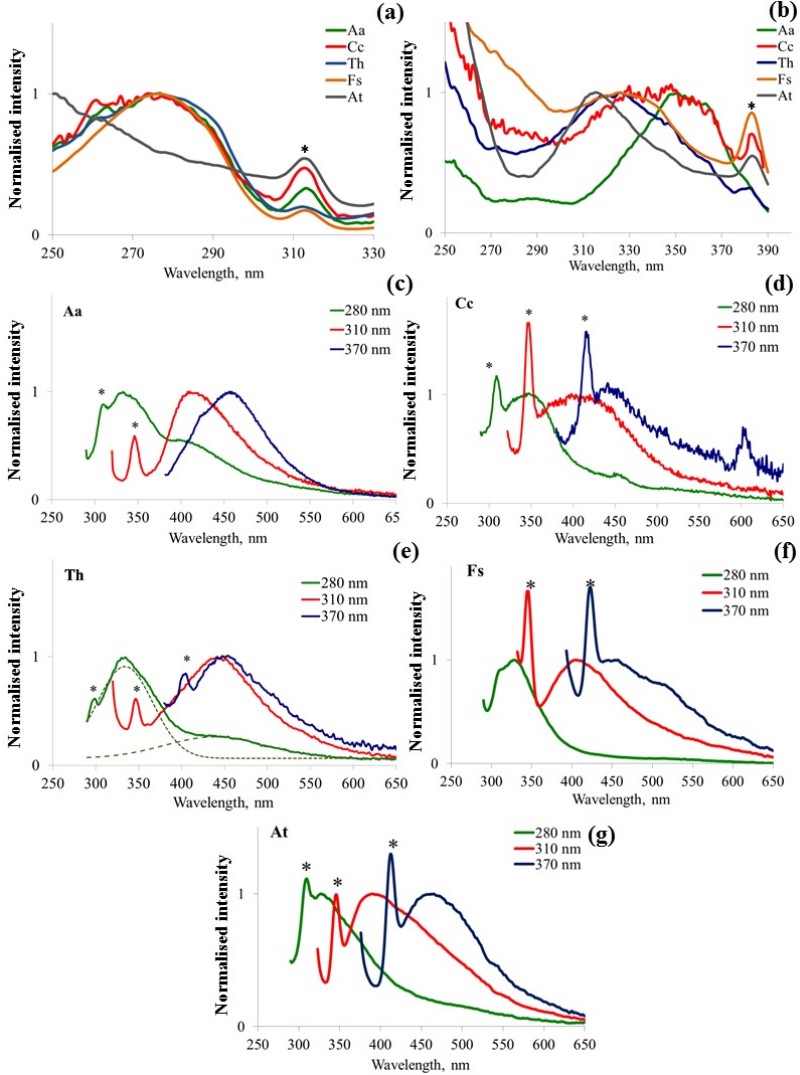

**Figure 3.** Fluorescence spectra of the fungal samples grown on agar medium: (**a**) excitation spectra with emission at 350 nm for *Alternaria alternata* (Aa), *Cladosporium cladosporioides* (Cc), *Trichoderma harzianum* (Th), *Fusarium solani* (Fs) and *Aspergillus terreus* (At); (**b**) excitation spectra with emission at 440 nm for Aa, Cc, Th, Fs and At; (**c**) emission spectra for Aa; (**d**) emission spectra for Cc; (**e**) emission spectra for Th; (**f**) emission spectra for Fs; and (**g**) emission spectra for At. The fluorescence emission spectra were measured with excitation at 280, 310 or 370 nm, and then normalized by emission intensity at maximum. The fluorescence excitation spectra were measured with registration of emission at 350 or 440 nm, then normalized by excitation intensity at maximum. The star indicates the position of water Raman scattering band.

Under excitation at 310 nm, the fluorescence spectra of fungal samples cultures differed in emission wavelength at maximum. The emission maximum of the blue fluorescence (400–410 nm) for *A. alternata, C. cladosporioides, F. solani* and A. *terreus* was found to be excitation wavelength-dependent, which supports the idea about the complex nature of their metabolites. These results resonate with data previously obtained for spectral analysis of humic substances (HS) [34]. The position of the emission maximum of natural aquatic HS depends on the excitation wavelength, λex: there is a shift of the maximum fluorescence to a range of short wavelengths (so-called 'blue shift' of the fluorescence) on changing the excitation wavelength from 270 to 310 nm, and the emission maximum of the band of humic-like fluorescence constantly shifts to the range of long wavelengths on increasing the excitation wavelength (λex $\geq$ 330 nm) [34]. Thus, the result obtained reveals that the spectral properties of fungal fluorophores (presumably melanin-like fluorophores) are similar to those of HS. The maximum emission with excitation at 280, 310 and 370 nm for *T. harzianum* were the same (Figure 3e), which could indicate the presence of NAD(P)H with more homogeneous structures than for other four fungi. The strains *A. alternata, C. cladosporioides, F. solani* and *A. terreus* also release NAD(P)H, the fluorescence emission of which overlaps with the emission of melanins [35,36].

With registration at 440 nm, we observed different peak locations in excitation spectra for the samples: at 350–370 nm for *A. alternata*; a broad peak at 320–370 nm for *C. cladosporioides*; the peak shifted to shorter wavelengths around 320–330 nm for *T. harzianum* and *F. solani*; and the peak shifted to shorter wavelengths around 310 nm for *A. terreus* (Figure 3b).

### 3.3. Fluorescence Spectral Features Analysis of Fungi Samples Cultivated in Liquid Medium

The fluorescence patterns of fungal samples cultivated as submerged cultures do not correspond completely to those for surface-associated cultures. Firstly, the main source of fluorophores in fungi cultivated on LM is extracellular metabolites, which contain more flavins. The differences for *T. harzianum* and *C. cladosporioides* samples are expressed by the fact that when cultivated in LM, a weak band with a maximum at 510–530 nm attributed to the emission of flavin compounds was recorded, while for samples cultured in AM, this band was less pronounced or was not recorded at all, respectively (Figures 3d and 4c; Figures 3e and 4d). For *F. solani* metabolites, the band at 510–560 nm was also recorded in both variants of cultivation. Secondly, in the samples *of A. alternata* grown in LM (Figure 4b), the broad band in the blue region with a maximum peaking at 400–500 nm became weaker compared to the samples grown on AM (Figure 3c). Conidial melanin pigmentation apparently is of some importance to tolerance to UV light, and also may provide survival advantages to these conidia [37]. According to [38], in *C. cladosporioides*, melanins are located in the walls of hyphae as well in conidia. Thus, strains of *A. alternata* and *C. cladosporioides*, spore suspensions of which revealed melanin fluorescence, can be considered as potentially more resistant strains compared to *T. harzianum* without melanins. In Supplementary Materials, Figures S2–S4 show the fluorescence spectra grouped differently: with the same cultivation method and with the same excitation wavelength.

### 3.4. PCA Analysis to Differentiate of Fungal Samples

The PCA of emission spectra was capable of meaningfully reducing the detection limit of microorganisms in model systems, as compared to the single ex/em wavelengths pair-based determination commonly used [25]. According to the PCA results, differences between fungal samples could be captured analyzing the fluorescence data at excitation at 280, 310 and 370 nm to varying degrees (Figure 5a–f). The first two principal components in fluorescence data fusion, accounting for 85–90% of the total data variance, were considered enough information to represent the entire samples [39]. Separation between a group of four fungal samples and *A. terreus* cultivated in aqueous medium was clearly observed analyzing the fluorescence data at excitation at 280 nm (Figure 5a,b). At the same time, the grouping of *A. alternata, C. cladosporioides, T. harzianum* and *F. solani* indicates the similarity

of protein-like fluorescence of fungal metabolites. As can be seen, the separation between fungal samples could be more differently observed based on fluorescence data at excitation at 310 nm which correspond to humic-like substances and melanin-like substances [32] (Figure 5c,d). The fungal samples were grouped in pairs *A. alternata—C. cladosporioides*, *F. solani—A. terreus* based on fluorescence at 310 nm data using PCA.

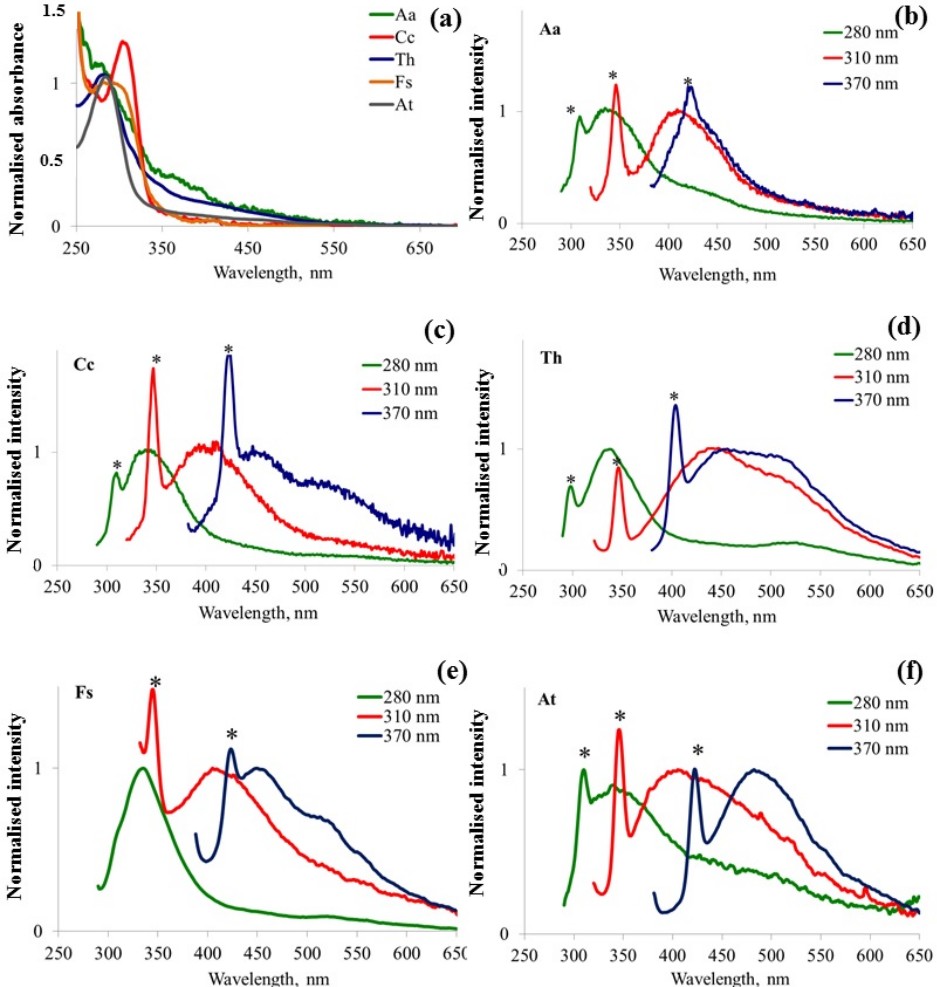

**Figure 4.** Absorption spectra normalized at absorbance taken at 280 nm (**a**) and fluorescence emission spectra of the fungal samples grown on liquid medium: (**b**) emission spectra of *Alternaria alternata* (Aa); (**c**) emission spectra of *Cladosporium cladosporioides* (Cc); (**d**) emission spectra of *Trichoderma harzianum* (Th); (**e**) emission spectra of *Fusarium solani* (Fs); and (**f**) emission spectra of *Aspergillus terreus* (At). The fluorescence emission spectra were measured with excitation at 280, 310 or 370 nm, and then normalized by emission intensity at maximum. The star indicates the position of water Raman scattering band.

Individual fungal samples were used for the PCA algorithm. In Supplementary Materials, Figure S7 shows distribution of all experimental replications of fungal samples on the PCA plot. Obtaining discrimination results for individual fungal samples is necessary to understand the patterns of their behavior in mixed samples. The next stage of work will be devoted to discrimination of individual fungal samples and their mixtures from bacterial samples and humic substances.

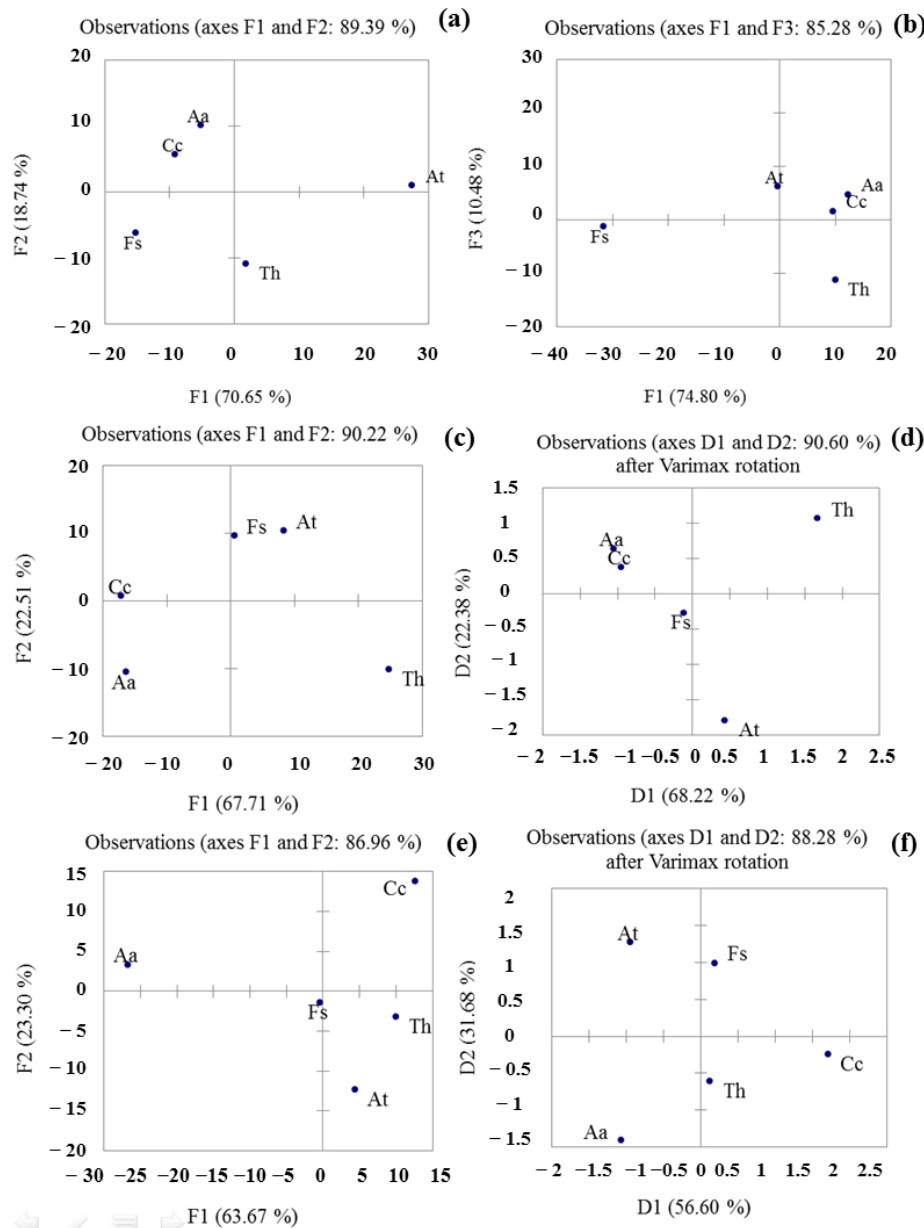

**Figure 5.** Discrimination of fungal samples with PCA: fluorescence data at excitation at 280 nm ((**a**) liquid medium; (**b**) agar medium); at 310 nm ((**c**) liquid medium; (**d**) agar medium), at 370 nm ((**e**)—liquid medium; and (**f**)—agar medium).

## 4. Discussion

The excitation spectra of protein fluorescence registered at 350 nm have been found to be identical for *A. alternata*, *C. cladosporioides*, *T. harzianum* and *F. solani* strains, except *A. terreus*, regardless of cultivation conditions (Figures 3, 4 and 5a–f). The QY values upon excitation at 270 nm (QY270) of most fungal samples (2.9, 2.9, 1.2, 2.8 and 2.4 for *A. alternata*, *C. cladosporioides*, *T. harzianum*, *F. solani* and *A. terreus*, respectively) are higher than the QY310 (0.8, 0.9, 0.7, 0.7 and 0.65 for *A. alternata*, *C. cladosporioides*, *T. harzianum*, *F. solani* and *A. terreus*, respectively) and QY355. This is due to the fact that the amount of protein, as one of the main primary metabolites, is undoubtedly higher in fungal culture liquids than in other DOM samples (natural and artificial HS) [40]. In addition, the intensity of protein fluorescence at 350 nm excited at 280 nm was found to correlate directly with the amount of spore biomass in water suspension. The Pearson correlation coefficients (R at $p = 0.05$) between data for conidia formation, in units of conidia per cm$^3$, and maxi-

mum fluorescence at 350 nm excited at 280 nm were 0.85 and 0.82 for *A. alternata* and *T. harzianum,* respectively [40]. These correlations were found for a quite large number of fungal samples grown on different variants of AM. Figure S6 in Supplementary Materials shows the graphs with regression equations. Biomass estimation is of crucial importance in biotechnology during a fermentation process and in ecology for estimation of pollution risk [2,7]. Tryptophan-like fluorescence excited at λex = 280–290 has been early used to estimate total biological fraction in environmental samples, fungal biomass in laboratory samples or to monitor bacterial contamination of water [25,41]. Therefore, the intensity of protein fluorescence can be used to approximately estimate the amount of fungal spores and biomass in the absence of other biological objects and to detect general microbial (bacterial or fungal) contamination.

It is important to discuss the blue emission with a maximum at 400–500 nm resembling the fluorescence of NAD(P)H, melanin-like compounds, or their mixture. According to previous studies, bacteria produce several types of melanin [42]. Moreover, bacterial fluorophores are not directly associated with melanin-like compounds. In particular, bacterial intrinsic fluorophores were divided into three regions: Region A (amino acids), Region N (NAD(P)H) and Region F (flavins) [28,39]. Fluorescence peak Ex/Em 360/450 representing HS was observed in the fungal system [8] and was not associated with humic-like and melanin-like components in bacteria [42]. Thus, the presence of excitation wavelength-dependent mode and the "blue shift" of fluorescence band of fungal samples can help to separate fungal and bacterial systems, as well as to distinguish fungal samples. Blue emission was more pronounced when fungal samples cultivated as surface-associated cultures.

This study indicates that fluorescence methods are helpful for the characterization of fungal fluorescence and the detection of fungal spores and biomass in water. Presence of other fluorophores, e.g., belonging to humic-like substances and bacterial metabolites, not related to fungal contamination, may obviously interfere with fluorescence-based analysis of fungal contamination of real water samples. The following ways seem to be beneficial in order to enhance fungal detection sensitivity: (i) utilizing sets of excitation–emission wavelengths combinations appropriate for detecting the emission of a wide range of fungal fluorophores; (ii) taking into account fungal fluorescence features, e.g., the presence of excitation wavelength-dependent mode and the "blue shift" of fluorescence band in different growth conditions; and (iii) fluorescence measurements (a single ex/em wavelengths pair and EEMs) have to be combined with PCA treatment (or with other statistical algorithms). Fluorescence methods can be suggested to enable fast and inexpensive monitoring of fungal contamination of such aquatic environments as coastal zones and wastewater discharge areas. Therefore, the approach using fluorescence is foremost recommended for pre-screening, monitoring and risk communication in water samples with low contents of natural organic matter such as groundwater [43].

## 5. Conclusions

Fungi are unique microorganisms playing a very important role in the biosphere and producing a wide range of metabolites. At the same time, fungal contamination of aquatic environments can lead to an adverse impact on the environment and human health, that initiates the actuality of fungi monitoring and detection. Our results demonstrate that the combination of fluorescence spectra obtained at a single ex/em wavelengths pair and whole EEMs coupled to PCA algorithms may offer the feasibility for early detection of potential risk of fungal contamination in aquatic environments water. The intrinsic fluorescence of fungi has not been studied so far, and our results of fluorescence studies performed on filamentous fungi with different pigmentation fill the gap.

**Supplementary Materials:** The following supporting information can be downloaded at: https://www.mdpi.com/article/10.3390/photonics9100692/s1. Figure S1. Fluorescence spectra (without normalization) of the fungal samples grown on agar medium: a—excitation spectra with emission at 350 nm for *Alternaria alternata* (Aa), *Cladosporium cladosporioides* (Cc), *Trichoderma harzianum* (Th), *Fusarium solani* (Fs), and *Aspergillus terreus* (At); b—excitation spectra with emission at 440 nm for Aa, Cc, Th, Fs, At; c—emission spectra for Aa; d—emission spectra for Cc; e—emission spectra for Th; f—emission spectra for Fs, g—emission spectra for At. Figure S2. Fluorescence spectra of the fungal samples grown on agar medium (AM) and liquid medium (LM): a—excitation spectra for *Alternaria alternata* (Aa), b—*Cladosporium cladosporioides* (Cc), c—*Trichoderma harzianum* (Th), d—*Aspergillus terreus* (At), and e—*Fusarium solani* (Fs). The fluorescence emission spectra were measured with excitation at 280, 310, or 370 nm, and then normalized by emission intensity at maximum of the fluorescence band. Figure S3. Fluorescence emission spectra of the fungal samples grown on agar medium: emission spectra with excitation at 280 (a), 310 (b), or 370 (c) nm for *Alternaria alternata* (Aa); *Cladosporium cladosporioides* (Cc); *Trichoderma harzianum* (Th); *Fusarium solani* (Fs), and *Aspergillus terreus* (At). The fluorescence emission spectra were normalized by emission intensity at maximum of the fluorescence band. The star indicates the position of water Raman scattering band. Figure S4. Fluorescence emission spectra of the fungal samples grown on liquid medium: emission spectra with excitation at 280 (a), 310 (b), or 370 (c) nm for *Alternaria alternata* (Aa); *Cladosporium cladosporioides* (Cc); *Trichoderma harzianum* (Th); *Fusarium solani* (Fs), and *Aspergillus terreus* (At). The fluorescence emission spectra were normalized by emission intensity at maximum of the fluorescence band. The star indicates the position of water Raman scattering band. Figure S5. Illustration of decomposition of the emission band at excitation at 280 gm for *Trichoderma harzianum* grown on a liquid medium into two Gaussians (a). Normalised fluorescence emission spectra of *Trichoderma harzianum* (Th) grown on liquid medium with Gaussian decomposition. Figure S6. Regression equations for fluorescence intensity registered at 350 nm with 280 nm excitation and sporulation intensity for a *Trichoderma harzianum* (a), b *Alternaria alternata* (b). Taken from (Fedoseeva et al., 2021) and modified. Figure S7. Discrimination of fungal samples with PCA: fluorescence data at excitation at 280 nm (a), at 310 nm (b).

**Author Contributions:** Conceptualization, E.F. and V.T.; methodology, S.P. and D.S.; software, S.P.; validation, E.F., S.P. and V.T.; formal analysis, E.F.; investigation, E.F. and S.P.; resources, E.F. and V.T.; data curation, D.S.; writing—original draft preparation, E.F.; writing—review and editing, S.P.; visualization, E.F.; supervision, D.S.; project administration, V.T.; funding acquisition, E.F. All authors have read and agreed to the published version of the manuscript.

**Funding:** The studies were funded by Russian Science Foundation (RSF) research project no. 22-24-00799.

**Institutional Review Board Statement:** Not applicable.

**Informed Consent Statement:** Not applicable.

**Data Availability Statement:** Not applicable.

**Acknowledgments:** The strains were kindly provided by Anna Ivanova from the collection of Department of Soil Biology, Faculty of Soil Science, Lomonosov Moscow State University. The authors sincerely thank Darya Khundzhua for performing some experiments and helping in visualizing the results.

**Conflicts of Interest:** The authors declare no conflict of interest.

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
