# Peer review of "Excitation-Dependent Fluorescence Helps to Indicate Fungal Contamination of Aquatic Environments and to Differentiate Filamentous Fungi"

_photonics, doi:10.3390/photonics9100692_

Round 1

Reviewer 1 Report

pls find the comments in the attached file

Author Response

Reviewer 1

The manuscript collected the EEMs of five kinds of fungi, and typically three spectra of each kind (under excitation of 280, 310, and 370 nm). PCA was employed to show the feasibility of classifying these five fungi. The intensity of protein fluorescence was claimed to be correlated with the biomass. The research target is not extensively investigated, as far as I know, but the manuscript needs much improvement before it can be published.

  1. Lack of relevant references. In line 61, the author claimed that fluorescence studies on fungi are few. If this is the case, then other methods, e.g., optical methods or chemical methods are required for comparison.

Thank you for this suggestion. We have extended our introduction section using the following references:

  1. Afonso T. B., Chaves Simões L.a, Lima N. Occurrence of filamentous fungi in drinking water: their role on fungal-bacterial biofilm formation. Res Microbiol. 2021 Jan-Feb;172(1):103791. doi: 10.1016/j.resmic.2020.11.002.
  2. Kadaifciler D. G., Demirel R. Fungal biodiversity and mycotoxigenic fungi in cooling-tower water systems in Istanbul, Turkey. J Water Health. 2017 Apr; 15(2):308-320. doi: 10.2166/wh.2017.274.
  3. Khundzhua, D.A.; Patsaeva, S.V.; Terekhova, V.A.; Yuzhakov V.I. Spectral characterization of fungal metabolites in aqueous medium with humus substances. Journal of Spectroscopy. 2013. Volume 2013. Article ID 538608. https://doi.org/10.1155/2013/538608
  4. Laptinskiy, K.A.; Burikov, S.A.; Patsaeva, S.V.; Vlasov, I.I.; Shenderova, O.A., Dolenko, T.A. Absolute luminescence quantum yield for nanosized carbon particles in water as a function of excitation wavelength. Spectrochimica Acta Part A: Molecular and Biomolecular Spectroscopy, 2020, Volume 229, 117879, https://www.sciencedirect.com/science/article/abs/pii/S1386142519312697
  5. Löbs Nina, Barbosa Cybelli G. G., Brill S., Walter D., Ditas F., Sá de Oliveira M., C. de Araújo A., R. de Oliveira L., Godoi R. H. M., Wolff S., Piepenbring M., Kesselmeier J., Artaxo P., Andreae M. O., Pöschl U., Pöhlker Ch., Weber B. Aerosol measurement methods to quantify spore emissions from fungi and cryptogamic covers in the Amazon. Atmos. Meas. Tech., 13, 153–164, 2020 https://doi.org/10.5194/amt-13-153-2020
  6. Al-gabr H., Zheng T., Yu X. Occurrence and quantification of fungi and detection of mycotoxigenic fungi in drinking water in Xiamen City, China Science of The Total Environment. 1 January 2014 Volumes 466–467 Pages 1103-1111
  7. Petra P. Identifying Fungi Spores, Yeast, Bacteria by Opto-Electronic Imaging and Image Processing and Identification for 0011 Protecting Human Health. Curr Trends Biomedical Eng & Biosci. 2018; 11(2): 555806. DOI: 10.19080/CTBEB.2018.11.555806
  8. Saari, S., J. Mensah‐Attipoe, T. Reponen, A. M. Veijalainen, A. Salmela, P. Pasanen, J. Keskinen. 2015. “Effects of fungal species, cultivation time, growth substrate, and air exposure velocity on the fluorescence properties of airborne fungal spores.” Indoor air 25 (6): 653-661. doi: 10.1111/ina.12166.
  9. Sarraguca, M.C., A. Paulo, M. M. Alves, A. M. A. Dias, J. A. Lopes, and E. C. Ferreira. 2009. “Quantitative monitoring of an activated sludge reactor using on-line UV–vis and near-infrared spectroscopy.” Anal. Bioanal. Chem. 395: 1159–1166. doi: 10.1007/s00216-009-3042-z.
  10. Schalk, R., D. Geoerg, J. Staubach, M. Raedle, F.-J. Methner, and T. Beuermann. 2017. “Evaluation of a newly developed mid-infrared sensor for real-time monitoring of yeast fermentations.” Journal of Bioscience and Bioengineering 123 (5): 651–657. doi: 10.1016/j.jbiosc.2016.12.005.
  11. Singh, G.P., S. Goh, M. Canzoneri, and R.J. Ram. 2015. “Raman spectroscopy of complex defined media. Biopharmaceutical applications.” Journal of Raman Spectroscopy 46: 545–550. doi: 10.1002/jrs.4686.
  12. Wang Y., Mao H., Xu G., Zhang X., Zhang Y. A Rapid detection method for fungal spores from greenhouse crops based on CMOS image sensors and diffraction fingerprint feature processing. J. Fungi 2022, 8(4), 374; https://doi.org/10.3390/jof8040374
  13. Zhiltsova, A.A.; Kharcheva, A.V.; Krasnova, E.D.; Lunina, O.N.; Voronov, D.A.; Savvichev, A.S.; Gorshkova, O.M.; Patsaeva, S.V.. Spectroscopic study of green sulfur bacteria in stratified water bodies of the Kandalaksha Gulf of the White Sea. Atmos Ocean Opt 31, 390–396 (2018). https://doi.org/10.1134/S1024856018040188

A number of laboratory techniques were developed for detection of fungal biomass and processes occurring with fungal participation in aquatic ecosystems. Among them one can find the microbiological cultural, molecular-genetics (PCR detection and sequencing) and chromatographic methods (Moh Al-gabr et al. 2014; Kadaifciler and Demirel, 2017; Afonso et al., 2020). Detection of fungi in aquatic ecosystems using traditional microbiological, molecular-genetics and chromatographic methods is species-specific, as well as time-consuming and instrument-complicated. Over the last few decades, there have been many investigations on fungi, aimed to monitor bioprocesses with the help of optical technologies, such as in situ microscopy, UV-visible, near-infrared and Raman spectroscopy. Optical methods are most widely used to detect airborne fungal particles (Saari et al., 2015; Petra, 2018; Löbs et al., 2020). Interesting image techniques have been developed for fungal spores monitoring: diffraction fingerprint image acquisition (Wang et al., 2022). Optical detection of fungi in aquatic environments due to its specifics has very limited applications, especially performed on mycelial forms of fungi (filamentous fungi) (Sarraguca et al. 2009; Khundzhua 2013  Singh et al. 2015; Schalk et al. 2017). It seems relevant to search for new methods to detect fungi in water, and spectral methods could provide us with fast and non-destructive methods in a non-contact mode. Fluorescence spectroscopy has been shown to be an important tool to characterize various OM in water sources (Laptinskiy, 2020). Fluorescence was linked to water quality indicators, such as pigments of anoxygenic phototrophic bacteria (Zhiltsova 2018), microbial biomass and, consequently, microbial contamination (Assawajaruwan et al., 2017; Nakar et al., 2019).

  1. The fluorescence emission spectra were measured with excitation radiation at three wavelengths, then EEM was also measured. Why there exists two methods to obtain fluorescence emission spectra? From EEM you can get the fluorescence emission spectra with excitation radiation at 280, 310 and 370 nm. What’s the difference between these two obtaining methods?

Measurement of EEM (emission-excitation matrix) is a more general approach, and of course includes measurement of emission spectra with excitation at three wavelengths like 280, 310 and 370 nm. The disadvantage of EEM measurements is their long duration, at least an hour for one sample. When measuring EEM, the researcher receives a lot of data, and not all of them are equally informative for the determination of fluorophores. A preliminary study of the spectra with different excitation wavelengths and analysis of the fluorescence excitation spectra led us to the conclusion that the most informative excitation wavelengths are 280, 310 and 370. Therefore, if the purpose of the study is express screening of the samples with on-line fluorescence analysis, then we recommend using emission registration with three excitation wavelengths. If the study is focused to a multilateral examination of the samples of unknown composition with non-limited time resource, then it is better to measure the EEM spectra, or, as they used to be called, the total luminescence spectra (TLS).

  1. What’s Fig. 2 (a, b) stand for? If Fig. 2(a) shows the spectra of the excitation of 280 nm, why the line for At is different from the other four? In line 131, the author claimed that the proteinaceous origin of the fluorophores emitted at 350 nm. However, it’s hard for me to find any obvious proof in Fig. 2(a). The description in line 131-133 cannot be deduced from Fig. 2(a).

If we understood correctly, then the question is why the fluorescence excitation spectra are shown when registering at 350 nm (Figure 2(a)) and 440 nm (Figure 2(b))?

The fluorescence excitation spectra recorded in the two main emission peaks are very important for understanding the nature of fungal fluorophores and their differences for different cultures. Figure 2(a) shows five curves, four of which are very similar, except for the relationship between the fluorescence intensity at maximum and the intensity of Raman scattering of water (marked with a star) reflecting the concentration of fluorophores.

We explain the nature of the fluorophores emitting in the UV range with maximum around 350 nm as protein-like or phenolic. This statement corresponds to the position of the maximum in the excitation spectrum at 280 nm. The fifth spectral curve (At) in this figure differs from the rest and it is difficult to identify a clear excitation maximum in the spectrum for it, since the intensity of protein-like fluorescence for this culture is much lower compared to that of all other samples. This is evidenced by the ratio of protein-like fluorescence bands and Raman bands in Figure 2(g).

In contrast to Figure 2(a) with similar in shape excitation spectra for protein-like fluorescence, all five curves in Figure 2(b) differ in shape when the emission spectrum is recorded at 440 nm. This indicates diversity of fluorophores in different fungal cultures, when emission is measured in the blue region.

  1. In line 201, the author claimed that the EEM is a non-expensive method. As far as I know, the EEM device is very expensive.

We do agree that modern spectroscopic equipment could be quite expensive. However, having the available luminescence spectrometer with alteration of excitation wavelengths, even with not very high spectral resolution (10 nanometers resolution is sufficient), it would be possible to apply EEM method to fungi fluorescent analysis. Because this analysis does not involve expensive sample treatment, the technique itself becomes not so expensive.

We corrected the statement about the EEM method in the text in the following way:

“In the recent years, EEM fluorescence spectroscopy being easy to implement, fast and relatively not so expensive method has been widely used …”

  1. The PCA method, though very commonly used, should be described in brief in Section 2. In Fig. 5, there seems only one sample of each kind of fungi was tested. The sample number is not big enough to support a paper. At least ten samples of each kind should be provided, otherwise, it is meaningless to use PCA, as they will be classified anyway if only one sample of each kind exist.

Thanks for the comment! Yes, we agree that along with an increase in the number of experimental data, the validity of any statistical analysis, including PCA, increases. The experiments were carried out in 3-5-fold biological replications. In Supplementary Materials, fig. 7S shows the distribution on the plot for all biological replications of the experimental samples. It can be seen that the samples are grouped in a similar way. For the 310 nm excitation data, Alternaria alternata and Cladosporium cladosporioides form a separate group, Fusarium solani and Aspergillus terreus also form a separate group, and Trichoderma harzianum is separated from others. As can be seen, the separation between fungal samples could be more differently observed based on fluorescence data at excitation at 310 nm which correspond to humic-like substances and melanin-like substances. For fluorescence excited at 280 nm, the samples do not form any defined groups. In the article, representative samples from all biological replications were selected for plotting the PCA to more clearly demonstrate the results. In other articles studied identification of different microorganism by fluorescence spectroscopy and classification through PCA, the authors use different numbers of samples (Giana et al., 2003; Mao et al., 2021). An important indicator is the account of total data variance. In our work, the first 2 principal components (factors) in fluorescence data fusion, accounted for 85-90% of the total data variance, were considered enough information to represent the entire samples (He et al., 2006; Gu et al., 2021).

Mao, Y., Chena, X-W., Chen, Z., Chena, G-Q., Lua, Y., Wu, Y-H., Hu, H-Y. Characterization of bacterial fluorescence: insight into rapid detection of bacteria in water. Water Reuse 2021, 11 (5). https://doi.org/10.2166/wrd.2021.040.

Giana, H.E., Silveira, L., Zângaro, R.A. and Pacheco, M.T.T. (2003) Rapid Identification of 587 Bacterial Species by Fluorescence Spectroscopy and Classification Through Principal 588 Components Analysis. Journal of Fluorescence 13(6), 489-493.

  1. Biomass prediction is also an important part of this paper, pls show the results in a figure or table, not just telling the correlation coefficients as line 254-259.

Yes, we agree that biomass prediction is also an important part of our study. The intensity of protein-like fluorescence at 350 nm was found correlating with the amount of conidia number in water suspension; larger is sporulation intensity, higher is fluorescence intensity. Though the dependence of fluorescence intensity was not linear versus conidia biomass, we found a good correlation between these two values. The Pearson correlation coefficients (at p=0.05) between data on conidia formation, in units of conidia per cm3, and maximum of fluorescence at 350 nm were 0.85 and 0.82 for A. alternaria and T. harzianum, respectively. Figure S6 in Supplementary Materials shows the graphs with regression equations. These correlations were received for a quite large number of fungi samples grown on different variants of nutrient media. The dependence of protein-like fluorescence intensity on the age and physiological state of fungal cultures makes it difficult to calculate the correlation. During the maturation of the T. harzianum cultures, conidia change their colour from yellow to green, and with an equal level of conidia formation, yellow-coloured conidia are less fluorescent at 350 nm (excited at 280 nm) than green-coloured conidia. Given such difficulties, we estimate as good the correlation of conidia biomass and fluorescence intensity measured at 350 nm with 280 nm excitation. These data are well described in our work - Elena V. Fedoseeva, Svetlana V. Patsaeva, Daria A. Khundzhua, Maria A. Pukalchik, Vera A. Terekhova. Effect of exogenic humic substances on various growth endpoints of Alternaria alternata and Trichoderma harzianum in the experimental conditions. Waste and Biomass Valorization. 2021. https://doi.org/10.1007/s12649-020-00974-x

  1. Pls read throughout the manuscript before submission. Some mistakes are very annoying, especially some residuals from the template, e.g., line 169 and line 324-334. In the first paragraph of Page 3, the unit gL-1 should be gL-1 ; line 256, the correlation coefficient should be represented as R, not R2, R2 is the coefficient of determination.

Corrected.

Author Response

Reviewer 2

Fungi play an important role in natural ecosystems. Fungi are commonly found in aquatic environments and can change human health as vectors through symbiotic relationships. Therefore, it is important to develop a rapid and inexpensive method to distinguish fungi in water. In this study, the authors used five fungi as models, using their internal fluorescent metabolites (e.g., amino acids, NAD(P)H, melanin-like compounds, and flavins), Excitation Emission matrices (EEMs) and Principal Component Analysis (PCA) algorithms are combined to identify five types of fungi under two different culture conditions. However, a major revision is required before publication. Specific comments are listed below.

  1. This study does not involve quantitative experimental data, so the word "quantification" mentioned many times in the paper (for example, Page 1, line26, Page 2, line 66, Page 8, line281) may lead readers to misunderstand that this work can detect fungi quantitatively.

Done. The word "quantification" was replaced by the words "monitoring" and "detection". If the context allowed, the word was removed.

  1. In the results section, it is hoped that the author can properly integrate the data in Fig. 2 and Fig. 3, so that readers can more intuitively observe the differences in fluorescence spectra of the five fungi under the two culture conditions.

We do not fully agree with this reviewer's proposal. The fact is that the series of illustrations in Figure 2 or in Figure 3 are grouped according to different methods of growing the fungi culture. Figure 2 shows the fluorescence excitation and emission spectra of the fungal samples grown on agar medium, and Figure 3 shows absorption spectra and fluorescence emission spectra of the fungal samples grown on liquid medium. Each Figure consists of 7 or 6 sections, and at least 3 spectral curves are drawn in each section. Combining these figures into one will lead to great confusion in the results of spectral measurements.

If, on the other hand, the reviewer's proposal concerns the combination of the same type of fluorescence spectra for different cultures (with the same cultivation method and with the same excitation wavelength), then such a comparison is added in the as Figure S2-S4 in the Supplementary materials.

  1. The reviewer would like to know whether the data used for the PCA algorithm are the fluorescence spectrum data of mixed samples from a variety of fungi, as the actual environment is co-existing with a variety of fungi. If not, then whether the mixed sample test can still get the desired discrimination results.

In the actual environment, a variety of fungi is co-existing with other fungi, bacteria and organic components. Presence of other fluorophores, e.g. belonging to humic-like substances and bacterial metabolites, not related to fungal contamination, may obviously interfere with fluorescence-based analysis of fungal contamination of real water samples. We used individual fungal samples for the PCA algorithm. Obtaining discrimination results for individual fungal samples is necessary to understand the patterns of their behavior in mixed samples. The next stage of work will be devoted to the discrimination of individual fungal samples and their mixtures from bacterial samples and humic substances.

We added in the Discussion:

«Individual fungal samples were used for the PCA algorithm. Obtaining discrimination results for individual fungal samples is necessary to understand the patterns of their behavior in mixed samples. The next stage of work will be devoted to discrimination of individual fungal samples and their mixtures from bacterial samples and humic substances»

  1. In line 124, the author indicated the spectral characteristics of five fungi in the aqueous environment, and why it is necessary to investigate their spectral characteristics on the solid surface first instead of directly analyzing the liquid cultured fungal samples.

In the aquatic environment, fungi can exist in two forms: planktonic cultures and surface-associated cultures. Planktonic cultures can be introduced into aquatic environments from land and soil. Surface-associated cultures may be involved in biofilm formation. The discovery of biofilm formation not only by bacteria and yeasts as well as filamentous fungi, has led to a better understanding of microbial ecology and to new insights into the mechanisms of virulence and persistence of pathogenic microorganisms (Harding et al. 2009; Siqueira et al. 2011). So, it seems very important to introduce more widely instrumental methods into the microbial ecology of aquatic ecosystems, in particular, spectral analysis (O'Connor et al. 2011; Saari et al. 2015). Analysis of the effect of organic substances, including contaminants, on the metabolic activity of the planktonic or surface-associated stages (spores and hyphal fragments) allows to distinguish the contribution of filamentous fungi in aquatic ecosystems and will more effectively select antifungal agents against opportunistic species and pathogens causing of human diseases.

  1. Page 4, Line 146-147," The maximum of emission with excitation at 280, 310, and 370 nm for T. Harzianum were the same (Fig. 2e) ", But from Fig. 2e, the maximum of emission with excitation at 280 nm is different from it at 310 and 370 nm. The reviewer would like the author to give an explanation.

Indeed, the green curve in Fig. 2e (fluorescence spectrum of the T. harzianum culture upon excitation at a wavelength of 280 nm) has the main maximum in the UV region. However, this fluorescence emission spectrum consists of two overlapping bands. The short-wavelength maximum corresponds to protein-like fluorescence; for this sample, it dominates the emission spectrum upon excitation at 280 nm. The second, less intense emission maximum is at the same wavelength as the fluorescence emission maximums upon excitation with two other wavelengths, 310 and 370 nm. We revised the statement in the text for a more correct one and changed Fig. 2e by adding components of the decomposition of the fluorescence band into two Gaussians (see also Supplementary materials). The deconvolution into Gaussian components was made only as illustration of the complexity of the fluorescence band excited at 280 nm, but not for quantitative alanysis.

  1. Page 3, Line 130, What is the meaning of “which additionally proves the proteinaceous origin of the fluorophores emitting at 350 nm”

Corrected in the text:

…which additionally proves the protein-like or phenolic-like origin of the fluorophores emitting at 350 nm.

  1. Page 5, Line 180-182. This information cannot be obtained from Fig. 3a. Is the data graph incorrectly referred to here? If not, it is hoped that the author can change a proper way to express themselves clearly.

This statement has removed.

  1. Page 6, Line 219, whether the literature has reported what exactly the "unique composition of Chromophores of A. terreus" is. If there is any such literature, please quote here as evidence.

  1. terreus is a producer of many aromatic secondary metabolites. The main secondary metabolites are various derivatives of asperlides and aspernolides, butyrolactones and butenolides, terrusnolides, which can potentially be the source of the unique composition of chromophores of A. terreus.

Jie Bao, Xiu-Xiu Li, Kongkai Zhu, Fei He, Yin-Yin Wang, Jin-Hai Yu, Xiaoyong Zhang, Hua Zhang. Bioactive aromatic butenolides from a mangrove sediment originated fungal species, Aspergillus terreus SCAU011. Fitoterapia Volume 150, April 2021, 104856. https://doi.org/10.1016/j.fitote.2021.104856

  1. In line 226, the author mentions the advantage of the low detection limit of PCA. Compared with the traditional method, how many times can EEM combined with PCA reduce the specific detection limit? The goal of the author is to realize the distinction between different fungi in the water environment, but the author only distinguished the spectral information of pure fungal samples. In the natural environment, most of them are a mixture of different fungi species. Under the spectral overlap and crosstalk, whether this method can still achieve the mixing of different species of fungi requires further experimental demonstration.

In the natural environment, a variety of fungi is co-existing with other fungi, bacteria and organic components. Presence of other fluorophores, e.g. belonging to humic-like substances and bacterial metabolites, not related to fungal contamination, may obviously interfere with fluorescence-based analysis of fungal contamination of real water samples. We used individual fungal samples for the PCA algorithm. Obtaining discrimination results for individual fungal samples is necessary to understand the patterns of their behavior in mixed samples. The next stage of work will be devoted to the discrimination of individual fungal samples and their mixtures from bacterial samples and humic substances.

We added in the Discussion:

«Individual fungal samples were used for the PCA algorithm. Obtaining discrimination results for individual fungal samples is necessary to understand the patterns of their behavior in mixed samples. The next stage of work will be devoted to discrimination of individual fungal samples and their mixtures from bacterial samples and humic substances»

  1. All of the References (page 10-11) should be cited in the format required by the journal.

Corrected.

  1. Page, 6, line 207, as described above “Fig. 4a-d” should be change to “Fig. 4a-e”.

Corrected.

  1. Page, 8, line 248, it should be “QY value” instead of “YQ value”.

Corrected.

  1. Page, 8, line 269-270, it should be “According to previous studies, bacteria produce several types of melanin[29]”.

Corrected.

Elena Fedoseeva.

Reviewer 3 Report

The aim of this study was to assess the autofluorescence spectroscopy, excitation-dependent fluorescence emissions, as a tool for a rapid evaluation of fungal biomass and to discriminate fungal samples. The ultimate goal of this research is to apply outcomes in environmental assessment and management. Overall the work contains a sufficient amount of autofluorescence spectroscopy, obtained at a single ex/em wavelengths pair and whole EEMs coupled to PCA algorithms of filamentous fungi, in order to assess the feasibility for early detection of potential risk of bacterial contamination in aquatic environments water. This work is of interest to a variety of people including those involved in environmental science, fluorescence and photonic community, etc.

Author Response

Reviewer 3

The aim of this study was to assess the autofluorescence spectroscopy, excitation-dependent fluorescence emissions, as a tool for a rapid evaluation of fungal biomass and to discriminate fungal samples. The ultimate goal of this research is to apply outcomes in environmental assessment and management. Overall the work contains a sufficient amount of autofluorescence spectroscopy, obtained at a single ex/em wavelengths pair and whole EEMs coupled to PCA algorithms of filamentous fungi, in order to assess the feasibility for early detection of potential risk of bacterial contamination in aquatic environments water. This work is of interest to a variety of people including those involved in environmental science, fluorescence and photonic community, etc.

There are no specific comments from the third reviewer, however we improved the introduction section and description of the results (Figures 2 and 3 and their description) for better understanding.

Elena Fedoseeva.

Round 2

Reviewer 1 Report

The manuscript is much improved and can be published along with the Supplementary Materials, especially, figure s6 and s7. My only concern to this version is the placement of Section 3.3. If the EEM only provides a big picture of where to excite and collect the fluorescence signals, along with the possible fluorophores, then may be the whole section should be placed at the beginning of Section 3, since it has nothing to do with the PCA process but supports the three wavelengths selection.

Author Response

Dear Reviewer,

We appreciate the valuable comment and have corrected our manuscript taking it into account.

The manuscript is much improved and can be published along with the Supplementary Materials, especially, figure s6 and s7. My only concern to this version is the placement of Section 3.3. If the EEM only provides a big picture of where to excite and collect the fluorescence signals, along with the possible fluorophores, then may be the whole section should be placed at the beginning of Section 3, since it has nothing to do with the PCA process but supports the three wavelengths selection.

Section “Fluorescence excitation/emission matrix (EEM) spectroscopy” was replaced at the beginning of Section 3.

Elena Fedoseeva

Reviewer 2 Report

 Accept in present form

Author Response

There are no specific comments.